# Social distancing is a social dilemma game played by every individual against his/her population

Zhijun Wu [1,2] *

**1** Graduate Program on Bioinformatics and Computational Biology, Iowa State University, Ames, Iowa, United States of America, **2** Department of Mathematics, Iowa State University, Ames, Iowa, United States of America

* zhijun@iastate.edu

**Citation:** Wu Z (2021) Social distancing is a social dilemma game played by every individual against his/her population. PLoS ONE 16(8): e0255543. https://doi.org/10.1371/journal.pone.0255543

**Data Availability Statement:** All relevant data are within the paper and its S1–S8 Files.

**Funding:** ZW receives support from the Simons Foundation on this work through the Mathematics and Physical Sciences Collaboration Grants for Mathematicians (Award Number: 586065). The funders had no role in study design, data collection

## Abstract

Since the outbreak of the global COVID-19 pandemic, social distancing has been known to everyone and recommended almost everywhere everyday. Social distancing has been and will be one of the most effective measures and sometimes, the only available one for fighting epidemics and saving lives. However, it has not been so clear how social distancing should be practiced or managed, especially when it comes to regulating everyone's otherwise normal social activities. The debate on how to implement social distancing often leads to a heated political argument, while research on the subject is lacking. This paper is to provide a theoretical basis for the understanding of the scientific nature of social distancing by considering it as a social dilemma game played by every individual against his/her population. From this perspective, every individual needs to make a decision on how to engage in social distancing, or risk being trapped into a dilemma either exposing to deadly diseases or getting no access to necessary social activities. As the players of the game, the individual's decisions depend on the population's actions and vice versa, and an optimal strategy can be found when the game reaches an equilibrium. The paper shows how an optimal strategy can be determined for a population with either closely related or completely separated social activities and with either single or multiple social groups, and how the collective behaviors of social distancing can be simulated by following every individual's actions as the distancing game progresses. The simulation results for populations of varying sizes and complexities are presented, which not only justify the choices of the strategies based on the theoretical analysis, but also demonstrate the convergence of the individual actions to an optimal distancing strategy in silico and possibly in natura as well, if every individual makes rational distancing decisions.

## Introduction

Since the outbreak of the global COVID-19 pandemic, social distancing has been known to everyone and practiced almost everywhere everyday. Social distancing has been and will be

and analysis, decision to publish, or preparation of
the manuscript.

**Competing interests:** The authors have declared
that no competing interests exist.

one of the most effective measures and sometimes, the only available one for fighting epidemics and saving lives [1–8]. Social distancing is not just simply keeping 6 feet away from each other. In a broader sense, it is to restrict regular social activities in whatever ways to avoid virus or bacteria infections, ranging from complete social isolation to limited business opening [9–12]. On one hand, complete isolation, although safe for health, is not socially or economically practical unless it is only for a few days or weeks. On the other hand, free participation in all social activities, although socially or economically beneficial, is dangerous of contracting deadly diseases. When in a severe pandemic, every individual needs to make a conscientious decision when managing his/her distancing practices [13–18].

Consider a simple case where there are only two types of activities to choose, one for quarantine such as staying home with only limited social activities such as going to grocery stores or gas stations, and another for more regular social activities including going to workplaces, schools, restaurants, etc. Then the question is what an individual in a given population should choose between the two types of activities? The answer often depends on what the whole population would do: If the whole population quarantines by choosing the first type of activities, the individual may instead take advantage to participate in the second type, where he/she can obtain more social or economic benefits yet without concerning his/her health because he/she may be the only one around there. However, if everyone else discovers this advantage, they will join this individual to get access to more social activities as well. The risk of contracting diseases is then increased for all. Everyone will retreat to the first type of activities for quarantine. The whole process then repeats due to the social dilemma every individual faces.

The above distancing behaviors can be described in a more concrete form as a social dilemma game between an individual and the population [19, 20]. Suppose the choices between the two types of activities correspond to two distancing strategies. Let's call the first type of activities as $Q$ for quarantine and the second type as $F$ for free-social. Then, there will be four extreme scenarios for the interactions between the individual and the population:

1. $(Q, F)$ / (quarantine, free-social);

2. $(Q, Q)$ / (quarantine, quarantine);

3. $(F, Q)$ / (free-social, quarantine);

4. $(F, F)$ / (free-social, free-social);

where each scenario is described by a pair of strategies, (Strategy $i$, Strategy $j$), with Strategy $i$ taken by the individual and Strategy $j$ by the population.

The four scenarios can be arranged into a $2 \times 2$ table as shown in Fig 1, with rows corresponding to the strategies of the individual and the columns to the strategies of the population. Each cell of the table corresponds to one of the scenarios. A number is assigned to each cell to represent an estimate on the amount of social contacts made by the individual in the corresponding interaction. In Scenario 1, the individual makes a small amount of social contacts, $\alpha > 0$. In Scenario 4, the individual makes a large amount of social contacts, $\beta > \alpha$. In Scenarios 2 and 3, the individual is assumed to make zero social contacts.

Assume that the goal of every individual is to minimize his/her social contacts while participating in either type of activities, quarantine ($Q$) or free-social ($F$). Suppose the game starts at Scenario 1, where the population is in a free-social state, but the individual chooses to quarantine. The individual makes zero social contacts. However, when everyone else realizes their risks of getting diseases, they will follow this individual, and the population will shift from free-social to quarantine. The game then goes to Scenario 2, where the individual and the population both choose to quarantine. The amount of social contacts the individual makes in this

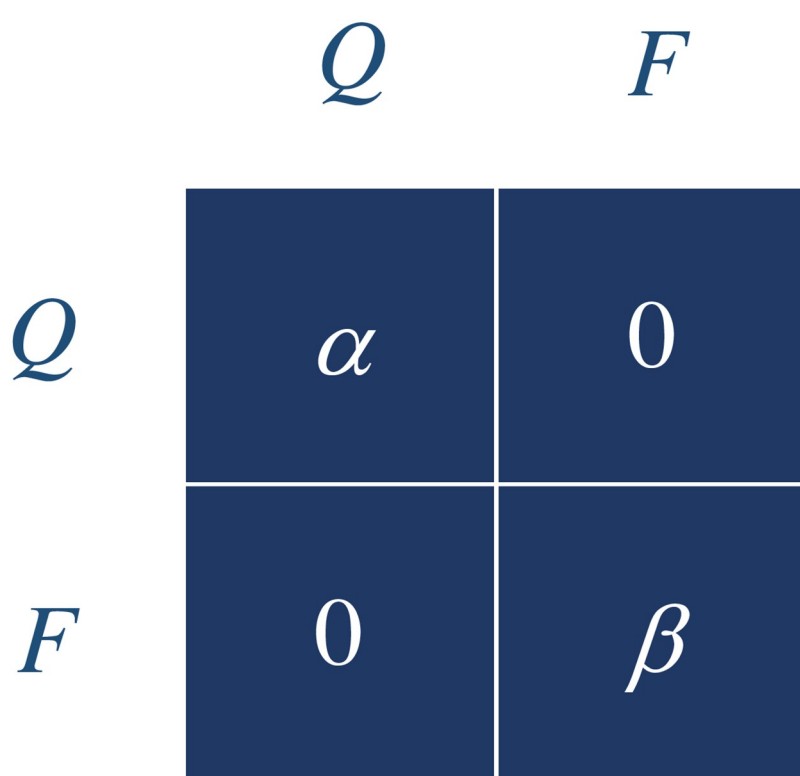

**Fig 1. Social distancing game.** A game with two choices of activities, either quarantine ($Q$) or free-social ($F$), can be described by a $2 \times 2$ table. The rows of the table correspond to the strategies played by an individual, while the columns to those by the population. A number is assigned to each cell to represent an estimate on the amount of social contacts the individual can make in the corresponding interaction. When the individual and population both choose to quarantine, the individual makes a small amount of social contacts $\alpha > 0$. When they both participate in free social activities, the individual makes a large amount of social contacts $\beta > \alpha$. In the other two cases, the individual makes zero social contacts.

state is then increased slightly to $\alpha$. However, the individual may change his/her strategy to free-social, for he/she can get access to more social activities without exposing to other individuals. The game then moves to Scenario 3, where the individual makes zero social contacts. It follows that after everyone else discovers this advantage, they will again join this individual for more social activities, and the whole population will shift from quarantine to free-social. The game then enters in Scenario 4, where the amount of social contacts of the individual is increased dramatically to $\beta$. In this state, to prevent from contracting or spreading diseases, the individual will immediately decide to change his/her strategy to quarantine. The game then returns to Scenario 1 again, and the whole process will repeat.

While in the cycle of scenarios $1 \to 2 \to 3 \to 4 \to 1$, the whole population will move back and forth between states where either all stay in quarantine or all have free social activities. However, in theory, there is a better strategy to avoid falling in this cycling trap. Instead of choosing complete quarantine or complete free-social, each individual can spend a certain part of his/her time for quarantine and the rest for free-social. Indeed, in the above simple game, if every individual spends $\beta/(\alpha + \beta)$ time for quarantine, while $\alpha/(\alpha + \beta)$ time for free-social, he/she will be able to reduce his/her social contacts to the least. Yet, any change in this strategy brings no incentive other than an increase in social contacts. The whole population then reaches an equilibrium with a probable fraction in quarantine and the rest free-social.

The latter situation has been observed in real world communities, where during the COVID-19 pandemic, social activities have not been completely closed, nor fully opened either.

A typical population of course has far more than two different types of social activities. The activities also have complex social connections among them. Then, an immediate question for the above game description of social distancing would be how optimal distancing strategies can be found for general cases. This question will be answered via further theoretical analysis on various types of distancing games. Results from computer simulation will also be presented to justify the theoretical analysis, and show that the optimal strategies can in fact be reached through the collective actions of the individuals in the population as well. However, a population may consist of different social groups such as students, working adults, seniors, etc. with different but partially shared social activities. Therefore, another question would be whether or not the game should be played collectively by every individual against only his/her subpopulation with a subset of social activities or strategies. The answer to this question is yes, and analysis and simulation on populations of different activity groups will also be discussed to demonstrate how the model can be extended from single to multiple populations.

## Results

### Optimal strategies: Balancing among multiple needs

Consider a general population with more than two, say $n$, different social activities. The activities may differ in the contents or locations, but may or may not be socially or physically completely separated, i.e., the participants in one activity may or may not have social contacts with those in any others. Assume that a contact value $\alpha_i \geq 1$ is assigned to each activity $i$. Let $x_i$ be the frequency (or probability) for an individual to participate in activity $i$, and $y_i$ the frequency (or probability) for the population to participate in activity $i$ in average. First, assume that the activities are independent of each other. Then it is easy to verify that the optimal distancing strategy for every individual is to choose a participating frequency $x_i^* = \lambda^*/\alpha_i$ for activity $i$, where $1/\lambda^* = 1/\alpha_1 + \cdots + 1/\alpha_n$. The distancing strategy of the whole population for activity $i$, i.e., $y_i^*$, then becomes $x_i^*$ as well, and the game reaches an equilibrium, with the amount of social contacts for every individual minimized to $\lambda^*$. Note that $x_i^*$ is inversely proportional to $\alpha_i$. Therefore, if the contact value $\alpha_i$ for activity $i$ is high, the optimal frequency $x_i^*$ to join this activity will be low (see Analysis General in Methods).

The activities may have some connections, where direct or indirect social contacts can be made by individuals in between. For example, individuals in a grocery store may come across those in a nearby cafeteria; individuals who go to two separate activities in different times but via the same bus line may have indirect contacts. However, if a so-called maximal independent set of activities can be found in the given activities, an optimal distancing strategy can still be determined by participating in only such a set of activities while avoiding all others. Let $\{1, \ldots, m\}$ represent the selected set of activities. Then, an optimal distancing strategy given $n$ social activities is to take $x_i^* = \lambda^*/\alpha_i$ for all $i = 1, \ldots, m$ and $x_i^* = 0$ for all $i = m + 1, \ldots, n$, if the maximum amount of social contacts one can make in activity $i = m + 1, \ldots, n$ is no less than $\lambda^*$, where $1/\lambda^* = 1/\alpha_1 + \cdots + 1/\alpha_m$. With this strategy, the amount of social contacts for every individual is again minimized to $\lambda^*$ at equilibrium (Analysis 1 in Methods).

A maximal independent set of activities is not the only choice for distancing. A dependent set of activities may also be selected if they form a maximal independent set of regular groups of activities. A regular group of activities is a group of activities, each having the same degree of connectivity, i.e., each connecting with the same number of other activities in the group. Consider a simple case where activities form connected pairs, but any two pairs are separated. A maximal set of such activities is called a maximal strong matching. Suppose there are $m$

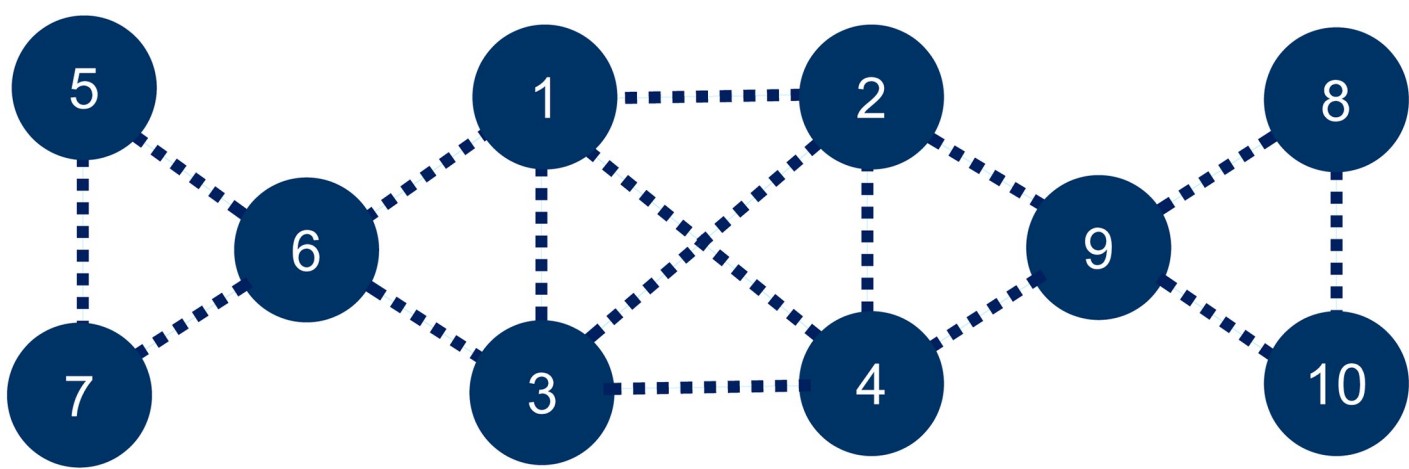

**Fig 2. Example population.** A population of ten possible activities, where activities {1, 2, 3, 4} are considered to be of high contact rates, and therefore assigned with a higher contact value $\alpha_i = 4$, $i = 1, \ldots, 4$. The rest of the activities are considered of low contact rates, and assigned with a lower contact value $\alpha_i = 1$, $i = 5, \ldots, 10$.

pairs of activities in a maximal strong matching. Let $\{i_1, i_2\}$ be the $i$th matched pair, $i = 1, \ldots,$ $m$. Assume the two activities in each matched pair are given the same contact value, i.e., $\alpha_{i_1} = \alpha_{i_2} = \alpha_i$. Then, it can be justified that an optimal distancing strategy can be obtained by choosing $\{x^*_{i_1}, x^*_{i_2}\}$ such that $x^*_{i_1} + x^*_{i_2} = \lambda^*/\alpha_i$ for all $i = 1, \ldots, m$, and $x^*_i = 0$ for all $i$ unmatched, if the maximum amount of social contacts one can make in an unmatched activity $i$ is no less than $\lambda^*$, where $1/\lambda^* = 1/\alpha_1 + \cdots + 1/\alpha_m$. The amount of social contacts for every individual is then minimized to $\lambda^*$ at equilibrium. Note that in general, the regular groups in the selected activities may have different group sizes and degrees. Let $V^*$ be the set of the selected activities, and $V^*_i$ be a connected regular group called a component in $V^*$, $i = 1, \ldots, m$. Let $m_i$ and $r_i$ be the size and degree of $V^*_i$, respectively, and assume that the activities in $V^*_i$ are given the same contact value $\alpha_i$. Then, an optimal distancing strategy can be obtained such that $x^*_j = 0$ for all $j$ not in $V^*$ and $\sum_{j \in V^*_i} x^*_j = \lambda^* m_i/\alpha_i/(r_i + 1)$ for all $i = 1, \ldots, m$, where $1/\lambda^* = \sum_{i=1}^m m_i/\alpha_i/(r_i + 1)$ (Analysis 2 in Methods).

Fig 2 shows an example population with ten possible activities. Assume that the activities {1, 2, 3, 4} are of high contact rates. They are therefore assigned with a higher contact value, say $\alpha_i = 4$, for all $i = 1, \ldots, 4$. The rest of the activities are of low contact rates, and are assigned with a lower contact value, say $\alpha_i = 1$, for all $i = 5, \ldots, 10$. The activities may or may not have connections. If they all are separated, an optimal distancing strategy can immediately be obtained with the frequencies $x^*_i = \lambda^*/\alpha_i = 1/28$ for all $i = 1, \ldots, 4$ and $x^*_i = \lambda^*/\alpha_i = 4/28$ for all $i = 5, \ldots, 10$, and the amount of social contacts is minimized to $\lambda^* = 1/7$ or about 14.29% for everyone while the maximum amount of social contacts that can be achieved is 4 or 400% (if the whole population goes to one of the first four activities).

If the activities do have connections as depicted with the dash lines in Fig 2, an optimal distancing strategy can still be obtained by selecting a maximal independent set of activities. For example, the activities {4, 6, 8} form a maximal independent set. Then, an optimal distancing strategy can be obtained with $x^*_4 = \lambda^*/\alpha_4 = 1/9$, $x^*_6 = \lambda^*/\alpha_6 = 4/9$, $x^*_8 = \lambda^*/\alpha_8 = 4/9$, and $x^*_i = 0$ for all $i \neq 4, 6, 8$. With this strategy, the amount of social contacts is then minimized to $\lambda^* = 4/9$ or about 44.44%. An optimal distancing strategy can also be obtained by selecting a dependent set of activities such as the activities {1, 2, 5, 7, 8, 10}, which form a maximal strong matching of the activities. The optimal strategy then becomes $x^*_1 + x^*_2 = \lambda^*/\alpha_1 = 1/9$,

$x_5^* + x_7^* = \lambda^*/\alpha_5 = 4/9$, $x_8^* + x_{10}^* = \lambda^*/\alpha_8 = 4/9$, and $x_3^* = x_4^* = x_6^* = x_9^* = 0$. With this strategy, the amount of social contacts is minimized to $\lambda^* = 4/9$ or about 44.44%, which is the same as the previous strategy, but it admits more activities, and allows flexible participating frequencies.

## Dynamic simulation: Collective behaviors of social distancing

The theoretical analysis on the distancing game can be justified through computer simulation if not experimental data which can be difficult to collect. With social distancing as the collective actions of the individuals in the population, the simulation is easy to implement just by following the changes of the individual strategies as they converge to an optimal one at equilibrium. The convergence is expected if every individual makes a rational distancing decision with the following deliberation: Every individual once a while examines the social contacts he/she may have in each activity; if the potential level of contacts in an activity is higher than the population average, the participating frequency for this activity is reduced; otherwise, it is increased. The successful convergence would also suggest that an optimal distancing strategy can be agreed upon naturally by the collective decisions repeatedly made by the individuals in the population (see Simulation General in Methods).

Consider the example population in Fig 2 again with all the activities assumed to be independent. Assume that there are 100 individuals in the population. A simulation was carried out for this population for 100 time periods, with each called a generation. At each generation, a change was made on every individual strategy once, followed by an update on the population strategy. In the end, almost all the individual strategies converged to an equilibrium one with $x_i^* \approx 0.0357 \approx 1/28$ for $i = 1, \ldots 4$ and $x_i^* \approx 0.1429 \approx 4/28$ for $i = 5, \ldots, 10$, and the social contact $\lambda^* \approx 0.1429 \approx 1/7$. These results agreed with all their theoretically expected values. They were also about the same in ten test runs with ten different sets of randomly generated initial strategies for all the individuals. Any subset of activities is an independent set, but the simulation never stopped with a partial completion on a subset of activities. It seemed to be able to expand the number of activities until a maximal independent set was reached, which includes all the activities for this population (Simulation 1 in Methods).

The dynamic behaviors of the above population in a typical simulation run are illustrated in Fig 3. The top two graphs show the initial and final distributions of the individual frequencies over the ten activities. The activities are listed along the $x$-axis. The individual frequencies are displayed in blue circles, one for one individual on one activity. The red stars represent the average frequencies of the population. At the initial stage, the frequencies were randomly distributed with the ranges varying from small to large, as shown in graph (A). In 100 generations, the frequencies for all the activities converged to their respective equilibrium ones, as shown in graph (B). In each generation, the root-mean-square-deviation (RMSD) of the activity frequencies of each individual deviated from the average frequencies of the population was calculated and then averaged over all the individuals. The blue curve in graph (C) shows the changes of the averaged RMSD over generations. The red curve is a smoother regression showing the general trend of the changes. As shown in this graph, the averaged RMSD gradually decreased to zero as the simulation progressed, further confirming the convergence of the individual frequencies to their equilibrium values (Simulation 1 in Methods).

Now assume that the activities do have some dependencies as depicted with the dash lines in Fig 2. Then, in theory, there are more than one equilibrium strategies, most notably, strategies on independent sets {6, 9}, {4, 6, 8}, {1, 5, 9}, and dependent sets {5, 6, 8, 9}, {6, 7, 9, 10}, {1, 3, 5, 7, 8, 10}, {2, 4, 5, 7, 8, 10}, {1, 3, 5, 7, 8, 9, 10}, and {2, 4, 5, 6, 7, 8, 10} as well. These strategies involve different numbers of activities and different degrees of connectivity, and therefore,

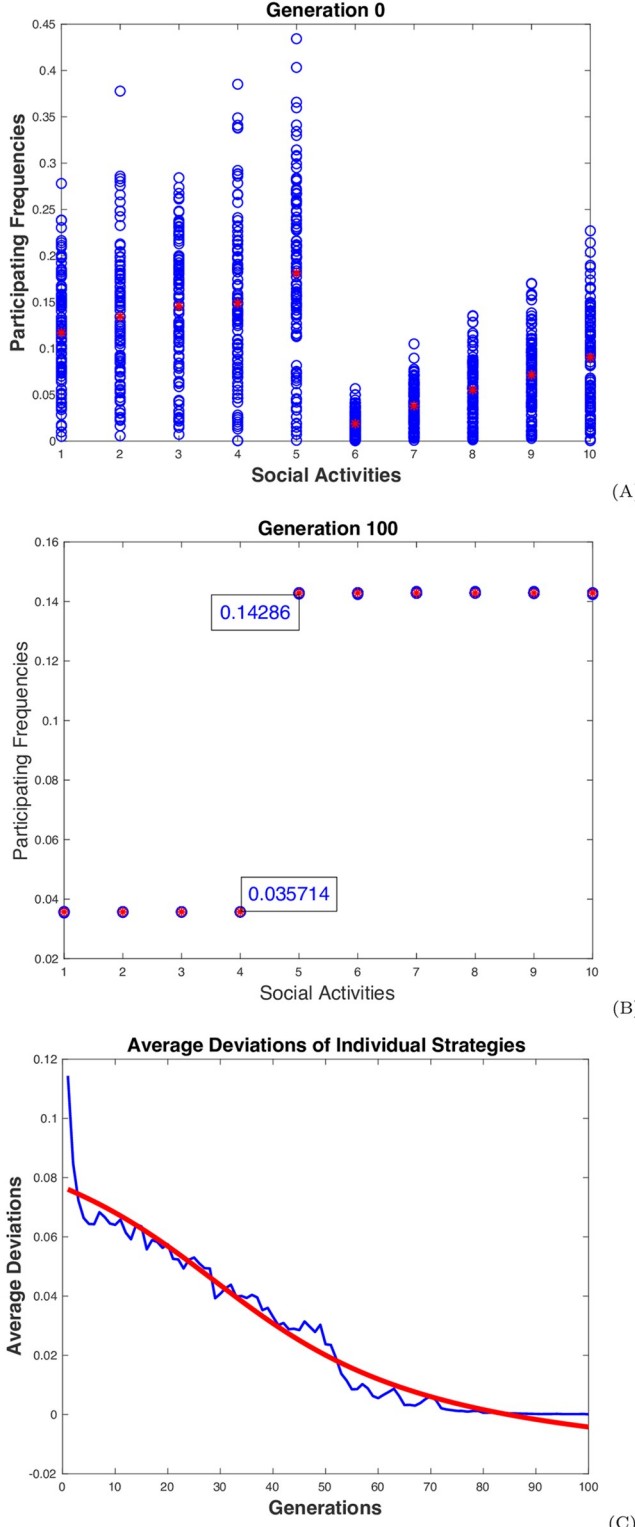

**Fig 3. Dynamic behaviors of the distancing population.** The simulation results from a typical test run for a population of 100 individuals with 10 independent activities are displayed: The graphs (A) and (B) show the initial and final distributions of the individual frequencies over the 10 activities. The activities are listed on the *x*-axis. The frequencies are displayed in blue circles, one for one individual on one activity. The red stars represent the average frequencies of the population. The blue curve in graph (C) shows the changes of the averaged RMSD of the individual frequencies over generations. The red curve is a smoother regression showing the general trend of the changes.

**Table 1. Simulation results for the population in Fig 2 with the assumed dependencies among the activities.**

| Act | Test 1 | Test 2 | Test 3 | Test 4 | Test 5 | Test 6 | Test 7 | Test 8 | Test 9 | Test 10 |
|---|---|---|---|---|---|---|---|---|---|---|
| 1 | 0.0924 | 0 | 0.0007 | 0 | 0.0474 | 0.0485 | 0.0384 | 0.0128 | 0.0340 | 0.0005 |
| 2 | 0 | 0.0915 | 0.0872 | 0.0584 | 0 | 0.0047 | 0 | 0.0320 | 0.0025 | 0.0373 |
| 3 | 0.0187 | 0 | 0.0006 | 0 | 0.0637 | 0.0526 | 0.0728 | 0.0151 | 0.0707 | 0.0013 |
| 4 | 0 | 0.0196 | 0.0226 | 0.0528 | 0 | 0.0053 | 0 | 0.0512 | 0.0039 | 0.0720 |
| 5 | 0.1880 | 0.1736 | 0.1599 | 0.1449 | 0.3331 | 0.3511 | 0.2083 | 0.2038 | 0.1953 | 0.1976 |
| 6 | 0 | 0.0289 | 0 | 0.0048 | 0 | 0 | 0 | 0 | 0 | 0 |
| 7 | 0.2564 | 0.2419 | 0.2845 | 0.2948 | 0.1113 | 0.0933 | 0.2361 | 0.2407 | 0.2492 | 0.2468 |
| 8 | 0.1966 | 0.1998 | 0.1968 | 0.1811 | 0.1786 | 0.1830 | 0.0950 | 0.3288 | 0.3402 | 0.2006 |
| 9 | 0.0053 | 0 | 0 | 0 | 0.0053 | 0 | 0.1177 | 0 | 0 | 0 |
| 10 | 0.2425 | 0.2447 | 0.2477 | 0.2634 | 0.2605 | 0.2614 | 0.2317 | 0.1157 | 0.1042 | 0.2439 |

Table Legends: Act—activities. Test #—simulation test. Each column—ending frequencies for corresponding activities.

have different levels of social contacts. However, between two strategies with different levels of social contacts, the one with lower social contacts would be preferred for obvious reasons. Also, between two strategies with the same level of social contacts but different numbers of activities, the one with more activities would be preferred, for it is socially or economically more beneficial (Simulation 2 in Methods).

A similar simulation to the previous one was run with the assumed dependencies among the activities for ten times with ten different sets of randomly generated initial strategies for all the individuals. The ending strategies were recorded as shown in Table 1, where the ending frequencies of the population on all the activities are listed for all ten test runs. The dynamic behaviors of the population in one of the simulation runs are illustrated in Fig 4. The top two graphs, (A) and (B), show the initial and final distributions of the individual frequencies over the ten activities. The convergence of the individual frequencies is demonstrated in graph (C), with the averaged RMSD of individual frequencies gradually decreasing to zero in 100 generations. In the ten test runs, the simulation converged to the following strategies: the one on {1, 3, 5, 7, 8, 9, 10} (3 times), with $x_1^* + x_3^* \approx 0.1111 \approx 1/9$, $x_5^* + x_7^* = x_8^* + x_9^* + x_{10}^* \approx 0.4444 \approx 4/9$, on {2, 4, 5, 6, 7, 8, 10} (2 times), with $x_2^* + x_4^* \approx 0.1111 \approx 1/9$, $x_5^* + x_6^* + x_7^* = x_8^* + x_{10}^* \approx 0.4444 \approx 4/9$, and on {1, 2, 3, 4, 5, 7, 8, 10} (5 times), with $x_1^* + x_2^* + x_3^* + x_4^* \approx 0.1111 \approx 1/9$, $x_5^* + x_7^* = x_8^* + x_{10}^* \approx 0.4444 \approx 4/9$. All these strategies selected a maximal independent set of regular groups of activities (not necessarily the same group size or degree), and their social contacts are all around $\lambda^* \approx 0.4444 \approx 4/9$ (Simulation 2 in Methods).

## Strategies for subpopulations: Do not have to act the same

In the real world, a population often consists of many different groups or in other words, subpopulations, such as students, essential workers, remote workers, seniors, etc. The subpopulations may have different but partially shared subsets of activities, and the individuals in different subpopulations may not take the same distancing strategy, i.e., there may not be a strategy optimal for everyone. However, the single population model discussed above can still be extended to multi-population cases, with the individuals playing the game against only their own subpopulations. Of course, the game in each subpopulation may depend on the outcomes of the games in other subpopulations if they share some activities. In terms of game theory, this is more like a multi-player game, with different players using different but partially shared subsets of strategies [19, 20].

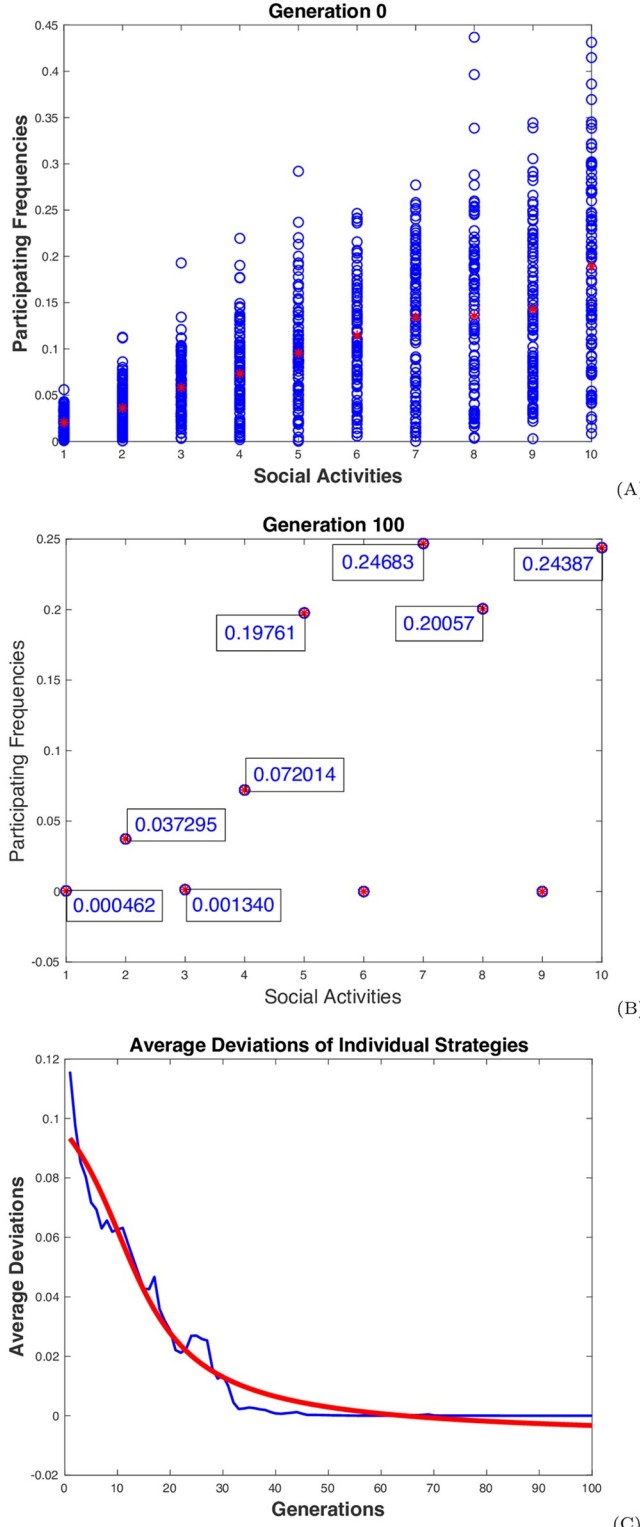

**Fig 4. Dynamic behaviors of the distancing population.** The simulation results from a typical test run for a population of 100 individuals with 10 dependent activities are displayed: The graphs (A) and (B) show the initial and final distributions of the individual frequencies over the 10 activities. The activities are listed on the *x*-axis. The frequencies are displayed in blue circles, one for one individual on one activity. The red stars represent the average frequencies of the population. The blue curve in graph (C) shows the changes of the averaged RMSD of the individual frequencies over generations. The red curve is a smoother regression showing the general trend of the changes.

Consider the case as described in Fig 2. Assume that the population is divided into two subpopulations called $P_1$ and $P_2$. The individuals in $P_1$ participate in activities $H_1 = \{1, 2, 3, 4, 5, 6, 7\}$, and those in $P_2$ participate in activities $H_2 = \{1, 2, 3, 4, 8, 9, 10\}$. Let $x_j^{(1)}$ be the frequency of participating in activity $j$ in $H_1$ by the individuals in $P_1$, $\sum_{j \in H_1} x_j^{(1)} = 1$, and $x_j^{(2)}$ the frequency of participating in activity $j$ in $H_2$ by the individuals in $P_2$, $\sum_{j \in H_2} x_j^{(2)} = 1$. If the activities are all separated, then it is not hard to show that an optimal distancing strategy is such that $x_j^{*(1)} = \lambda^*/\alpha_j$ for all $j$ in $H_1 \setminus H_2$, $x_j^{*(2)} = \lambda^*/\alpha_j$ for all $j$ in $H_2 \setminus H_1$, and $x_j^{*(1)} + x_j^{*(2)} = \lambda^*/\alpha_j$ for all $j$ in $H_1 \cap H_2$, which minimizes everyone's social contacts to $\lambda^*$, where $1/\lambda^* = (1/\alpha_1 + \cdots + 1/\alpha_n)/2$. If the activities are connected as shown in Fig 2, assume a maximal strong matching of activities is selected, and assume every matched pair of activities $\{j_1, j_2\}$ is either entirely in $H_1$ or in $H_2$ and has the same contact value $\alpha_j$. Then, $x_j^{*(1)} = x_j^{*(2)} = 0$ for all $j \in H_1 \cup H_2$ not matched, $x_{j_1}^{*(1)} + x_{j_2}^{*(1)} = \lambda^*/\alpha_j$ for all matched pair $\{j_1, j_2\}$ in $H_1 \setminus H_2$, $x_{j_1}^{*(2)} + x_{j_2}^{*(2)} = \lambda^*/\alpha_j$ for all matched pair $\{j_1, j_2\}$ in $H_2 \setminus H_1$, and $x_{j_1}^{*(1)} + x_{j_2}^{*(1)} + x_{j_1}^{*(2)} + x_{j_2}^{*(2)} = \lambda^*/\alpha_j$ for all matched pair $\{j_1, j_2\}$ in $H_1 \cap H_2$, where $1/\lambda^* = (1/\alpha_1 + \cdots + 1/\alpha_m)/2$. Note that in general, the selected activities may be a maximal independent set of regular groups of different sizes and degrees (Analysis 3 in Methods).

The above analysis was also confirmed by computer simulation. The simulation was run for ten times with ten different sets of randomly generated initial strategies for all the individuals. Tables 2 and 3 contain the results from the simulation with the dependencies among the activities as shown in Fig 2. The ending strategies for subpopulation $P_1$ are listed in Table 2 and for subpopulation $P_2$ in Table 3. They all agree with their expected equilibrium values. For example, in Test 1, a set of activities, $\{1, 3, 5, 7, 8, 9, 10\}$, was selected for distancing by the simulation. For the selected activities in $H_1 \setminus H_2$, i.e., $\{5, 7\}$, $x_5^{*(1)} + x_7^{*(1)} \approx 0.3902 + 0.4987 = 0.8889 \approx 8/9 = \lambda^*/\alpha_5$. For the selected activities in $H_2 \setminus H_1$, i.e., $\{8, 9, 10\}$, $x_8^{*(2)} + x_9^{*(2)} + x_{10}^{*(2)} = 0.3823 + 0.0453 + 0.4614 = 0.8890 \approx 8/9 = \lambda^*/\alpha_8$. For the selected activities in $H_1 \cap H_2$, i.e., $\{1, 3\}$, $x_1^{*(1)} + x_3^{*(1)} + x_1^{*(2)} + x_3^{*(2)} \approx 0.0972 + 0.0140 + 0.0914 + 0.0197 = 0.2223 \approx 2/9 = \lambda^*/\alpha_1$, where $\lambda^* = 8/9$ (Simulation 3 in Methods).

The dynamic behaviors of single or multiple subpopulations demonstrated above are also observed when the dependencies among the activities in Fig 2 are changed. The simulation

**Table 2. Simulation results for the population in Fig 2 with two subpopulations $P_1$ and $P_2$, each having a preferred subset of activities.**

| Act | Test 1 | Test 2 | Test 3 | Test 4 | Test 5 | Test 6 | Test 7 | Test 8 | Test 9 | Test 10 |
|---|---|---|---|---|---|---|---|---|---|---|
| 1 | 0.0972 | 0.0500 | 0 | 0 | 0.0385 | 0.0022 | 0.0524 | 0 | 0.0364 | 0 |
| 2 | 0 | 0.0498 | 0.1071 | 0.0646 | 0 | 0.0466 | 0 | 0.0368 | 0 | 0.0321 |
| 3 | 0.0140 | 0.0035 | 0 | 0 | 0.0726 | 0.0014 | 0.0587 | 0 | 0.0747 | 0 |
| 4 | 0 | 0.0077 | 0.0040 | 0.0465 | 0 | 0.0610 | 0 | 0.0743 | 0 | 0.0790 |
| 5 | 0.3902 | 0.3732 | 0.3603 | 0.3150 | 0.6049 | 0.5849 | 0.4550 | 0.4176 | 0.3438 | 0.3643 |
| 6 | 0 | 0 | 0 | 0.0204 | 0 | 0 | 0 | 0.0035 | 0 | 0.0219 |
| 7 | 0.4987 | 0.5157 | 0.5286 | 0.5535 | 0.2840 | 0.3040 | 0.4339 | 0.4677 | 0.5451 | 0.5026 |
| 8 | 0 | 0 | 0 | 0 | 0 | 0 | 0 | 0 | 0 | 0 |
| 9 | 0 | 0 | 0 | 0 | 0 | 0 | 0 | 0 | 0 | 0 |
| 10 | 0 | 0 | 0 | 0 | 0 | 0 | 0 | 0 | 0 | 0 |

Table Legends: Act—activities. Test #—simulation test. Each column—ending frequencies for corresponding activities for $P_1$.

**Table 3. Simulation results for the population in Fig 2 with two subpopulations $P_1$ and $P_2$, each having a preferred subset of activities.**

| Act | Test 1 | Test 2 | Test 3 | Test 4 | Test 5 | Test 6 | Test 7 | Test 8 | Test 9 | Test 10 |
|-----|--------|--------|--------|--------|--------|--------|--------|--------|--------|---------|
| 1 | 0.0914 | 0.0786 | 0 | 0 | 0.0589 | 0.0001 | 0.0366 | 0 | 0.0271 | 0 |
| 2 | 0 | 0.0179 | 0.0960 | 0.0577 | 0 | 0.0577 | 0 | 0.0429 | 0 | 0.0368 |
| 3 | 0.0197 | 0.0119 | 0 | 0 | 0.0522 | 0.0009 | 0.0745 | 0 | 0.0840 | 0 |
| 4 | 0 | 0.0027 | 0.0151 | 0.0534 | 0 | 0.0524 | 0 | 0.0683 | 0 | 0.0743 |
| 5 | 0 | 0 | 0 | 0 | 0 | 0 | 0 | 0 | 0 | 0 |
| 6 | 0 | 0 | 0 | 0 | 0 | 0 | 0 | 0 | 0 | 0 |
| 7 | 0 | 0 | 0 | 0 | 0 | 0 | 0 | 0 | 0 | 0 |
| 8 | 0.3823 | 0.4271 | 0.3965 | 0.4379 | 0.4266 | 0.3516 | 0.2656 | 0.6267 | 0.7237 | 0.4280 |
| 9 | 0.0453 | 0 | 0 | 0 | 0.0253 | 0 | 0.1111 | 0 | 0.0293 | 0 |
| 10 | 0.4614 | 0.4618 | 0.4924 | 0.4510 | 0.4370 | 0.5373 | 0.5122 | 0.2621 | 0.1359 | 0.4609 |

Table Legends: Act—activities. Test #—simulation test. Each column—ending frequencies for corresponding activities for $P_2$.

results are obtained and stored in S1 and S2 Files for either single or multiple subpopulations with 10 social activities to form a Petersen's diagram.

## Beyond model problems: A more realistic example

A simulation was also carried out for a larger population in a more complex social environment resembling a small university town, Ames, Iowa, where Iowa State University is located. The social activities in the town can be divided into 14 groups with roughly 85 different activities at different locations (see Fig 5 and description below). For each activity, a contact value can be estimated according to the average physical proximity of the person-to-person interactions in the activity. However, some activities are more essential than others and have higher participating frequencies even if their contact values are high. Likewise, some are less essential than others and have lower participating frequencies even if their contact values are low. To take such a factor into account, the estimated contact values are further adjusted based on the importance of the activities: If $c_i$ is the estimated contact value of activity $i$, then a contact value $\alpha_i = c_i/s_i$ is actually assigned to activity $i$, where $s_i$ is an estimate on the importance of the activity, called the importance factor, $1/2 \leq s_i \leq 2$. If $1 < s_i \leq 2$, activity $i$ is considered to be more important or essential than average, and its contact value is downgraded. If $1/2 \leq s_i < 1$, activity $i$ is considered to be less important or essential, and its contact value is then upgraded. Use $c_i \geq 2$. Then, $\alpha_i \geq 1$.

- **College Campus**: Activities 1–8, including classrooms, office buildings, cafeterias, student activity centers, surrounding labs, service buildings, organizations. Estimated contact values: 8; importance factors: 4/3; assigned contact values: 6.

- **College Town**: Activities 9–14, including pharmacies, banks, printing services, post offices, barbershops, restaurants, student dorms. Estimated contact values: 8; importance factors: 4/3; assigned contact values: 6.

- **W. HyVee**: Activities 15–24 at the west HyVee plaza, including grocery stores, clinics, restaurants, banks, city offices, fitness centers, beauty shops. Estimated contact values: 4; importance factors: 2; assigned contact values: 2.

- **Somerset**: Activities 25–28 at a small community center, including restaurants, fitness centers, business offices, flower shops, salons, clinics, banks. Estimated contact values: 4; importance factors: 1; assigned contact values: 4.

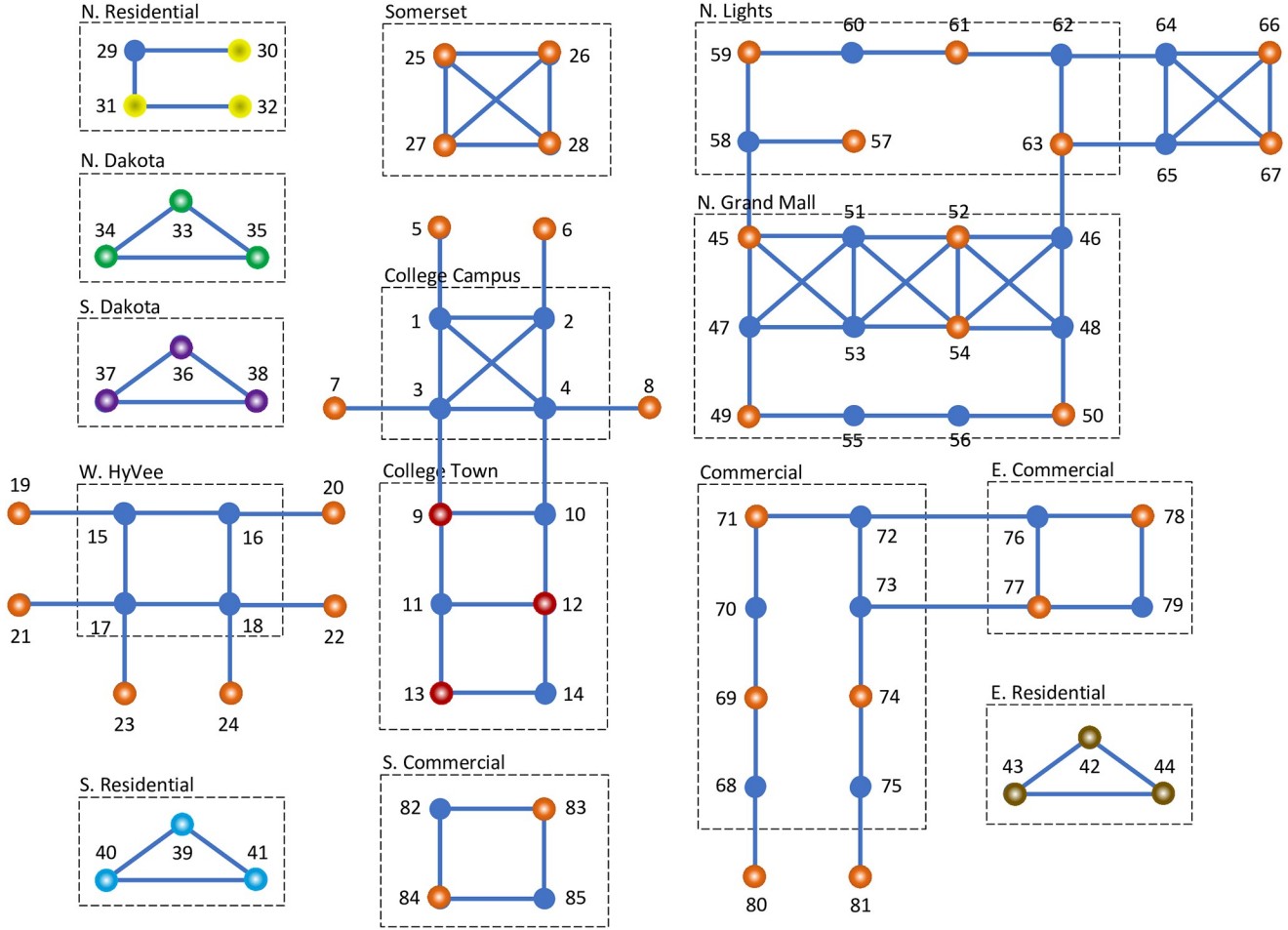

**Fig 5. Simulation of social distancing in a university town.** Nodes represent social activities. Links represent contact connections. The population is divided into 6 subpopulations, $P_1, \ldots, P_6$, each having a preferred subset of social activities. In the end of the simulation, a maximal independent set of regular groups of activities is selected by the subpopulations together. Those colored in dark blue are not selected by any subpopulation. Those colored in red are selected by $P_1$, yellow by $P_2$, green by $P_3$, purple by $P_4$, light blue by $P_5$, brown by $P_6$, and orange by more than one subpopulations.

- **N. Residential**: Activities 29–32 in the northern residential area, including home activities, nearby convenient stores, and gas stations. Estimated contact values: 2; importance factors: 2; assigned contact values: 1.

- **N. Dakota**: Activities 33–35 in the North Dakota residential area, including home activities, nearby convenient stores, and gas stations. Estimated contact values: 2; importance factors: 2; assigned contact values: 1.

- **S. Dakota**: Activities 36–38 in the South Dakota residential area, including home activities, nearby convenient stores, and gas stations. Estimated contact values: 2; importance factors: 2; assigned contact values: 1.

- **S. Residential**: Activities 39–41 in the southern residential area, including home activities, nearby convenient stores, and gas stations. Estimated contact values: 2; importance factors: 2; assigned contact values: 1.

- **E. Residential**: Activities 42–44 in the east residential area, including home activities, nearby convenient stores, and gas stations. Estimated contact values: 2; importance factors: 2; assigned contact values: 1.

- **N. Grand Mall**: Activities 45–56 in the northern mall area. Inside the Mall, there are about 20 retail stores, a couple of small restaurants, and a movie theater. Outside the Mall, there are several surrounding business buildings including drug stores, banks, and restaurants. Estimated contact values: 4 for activities 45–48 and 51–54, and 2 for activities 49, 50, 55, 56. The importance factors are all given 1, and therefore, the assigned contact values are all the same as the estimated ones.

- **N. Lights**: Activities 57–67 in a northern business plaza called the Northern Lights, including restaurants, food stores, beauty shops, gas stations, and a Walmart superstore. Estimated contact values: 2 for activities 57–63, and 4 for activities 64–67. Again the importance factors are all given 1, and therefore, the assigned contact values are all the same as the estimated ones.

- **Commercial**: Activities 68–75 and 80–81, including all in the old town and along a newly developed commercial street, the most busy and crowded area in town. Estimated contact values: 4; importance factors: 1; assigned contact values: 4.

- **E. Commercial**: Activities 76–79, in the east part of the town, including hotels, car garages, material retailers, small factories, car dealers. Estimated contact values: 4; importance factors: 2; assigned contact values: 2.

- **S. Commercial**: Activities 82–85, in the southern part of the town, including whole sale stores, home improvement stores, car dealers, research labs, tech companies. Estimated contact values: 4; importance factors: 1; assigned contact values: 4.

Note that in order to simplify the model, the number of activity nodes in each of the above areas is much smaller than the actual number, but they are representative of the activities there. For example, there is a circle of only three activity nodes in most residential areas, which could be expanded to a much larger circle of many neighboring residential blocks, but the home-gas-store triangle may be adequate to describe the major activities in the areas, and the nature of the model is not changed much.

The population is divided into 6 different groups based on their residential locations. Each group has its own preferred activities, some exclusively for the group and some shared with other groups:

- $P_1$ **subpopulation**: All students living around College Town. Their preferred activities include all in College Town, College Campus, W. HyVee, and N. Grand Mall.

- $P_2$ **subpopulation**: All residents in N. Residential. Their preferred activities include all in N. Residential, College Campus, Somerset, N. Lights, N. Grand Mall, Commercial, E. Commercial, and S. Commercial.

- $P_3$ **subpopulation**: All residents in N. Dakota. Their preferred activities include all in N. Dakota, College Campus, W. HyVee, Somerset, N. Grand Mall, Commercial, E. Commercial, and S. Commercial.

- $P_4$ **subpopulation**: All residents in S. Dakota. Their preferred activities include all in S. Dakota, College Campus, W. HyVee, Somerset, N. Grand Mall, Commercial, E. Commercial, and S. Commercial.

- **$P_5$ subpopulation**: All residents in S. Residential. Their preferred activities include all in S. Residential, College Campus, W. HyVee, Somerset, N. Grand Mall, Commercial, E. Commercial, and S. Commercial.

- **$P_6$ subpopulation**: All residents in E. Residential. Their preferred activities include all in E. Residential, College Campus, Somerset, N. Lights, N. Grand Mall, Commercial, E. Commercial, and S. Commercial.

The whole population is assumed to have 850 individuals, with 250 in $P_1$, 120 in each of $P_2$, $P_3$, $P_4$, $P_5$, and $P_6$. The actual population size is about 100 times larger, but for simulation, a small population size is assumed for more efficient testings.

The simulation was carried out for 10 times with 10 different randomly generated initial strategies for all the individuals in the population. In each time, it ran for 400 generations, and in the end, converged successfully to an equilibrium state where each subpopulation reached its own optimal distancing strategy. All simulation results were recorded and documented in Supporting information. The activities selected by the ending strategies in all ten simulation runs were almost the same as those color-coded in Fig 5: The locations in dark blue are those not selected by any subpopulation at equilibrium. All other colored locations are those selected by some subpopulations with certain non-zero participating frequencies. The activities colored in red are those selected exclusively by subpopulation $P_1$, yellow by $P_2$, green by $P_3$, purple by $P_4$, light blue by $P_5$, and brown by $P_6$. The activities colored in orange are those shared and selected by more than one subpopulations.

As shown clearly in Fig 5, the selected activities form a maximal independent set of regular groups. This means that no activities were left unselected yet separated from the selected ones or in other words, each unselected activity has at least one connection with the selected activities. In addition, each connected component of the selected activities is a regular group of activities with the degree ranging from 0 to 3. Let $V_i^*$ be the set of activities in the $i$th connected component of the selected activities, $m_i = |V_i^*|$. Let $r_i$ and $\alpha_i$ be the degree and the contact value of the activities in $V_i^*$, respectively. Then, based on the previous analysis, at equilibrium, the average amount of social contacts minimized for all the subpopulations is equal to $\lambda^*$ such that $1/\lambda^* = \left(\sum_{i=1}^m m_i/\alpha_i/(r_i + 1)\right)/M$, where $m$ is the number of connected components in the selected activities and $M$ is the number of subpopulations. For the strategy shown in Fig 5, $m = 38$, and $M = 6$. Therefore, it is easy to verify that $\lambda^* = 72/203$ (Analysis 3 in Methods).

Let $x_j^{*(k)}$ be the optimal frequency for the subpopulation $P_k$ to participate in activity $j \in V_i^*$. Then, $\sum_{k:V_i^* \subset H_k} \sum_{j \in V_i^*} x_j^{*(k)} = \lambda^* m_i/\alpha_i/(r_i + 1)$. For example, the activity 30 in N. Residential is a component of the selected activities with only one activity and contact value equal to 1. It is also selected only by subpopulation $P_2$. Therefore, $x_{30}^{*(2)} = 72/203$. The regular group of activities 33, 34, and 35 in N. Dakota is a component of the selected activities with contact value equal to 1. It is also selected only by subpopulation $P_3$. Therefore, $x_{33}^{*(3)} + x_{34}^{*(3)} + x_{35}^{*(3)} = 72/203$. Also, the activity 57 in N. Lights is a component of the selected activities with only one activity and contact value equal to 2. However, it is shared by two subpopulations $P_2$ and $P_6$. Therefore, $x_{57}^{*(2)} + x_{57}^{*(6)} = 36/203$. For the selected activities, the simulated frequencies all converged to their theoretical values. For example, $x_{30}^{*(2)} \approx 0.354680 \approx 72/203$, $x_{33}^{*(3)} + x_{34}^{*(3)} + x_{35}^{*(3)} \approx 0.113215 + 0.118777 + 0.122688 = 0.354680 \approx 72/203$, and $x_{57}^{*(2)} + x_{57}^{*(6)} \approx 0.043988 + 0.133352 = 0.177340 \approx 36/203$. (Simulation 3 in Methods).

Note that the simulation was also run without dividing the population, i.e., no subsets of activities for subpopulations, and the individuals can choose among all 85 activities. The activities selected at equilibrium were almost the same as in the multi-population case, i.e., the

population also selected a maximal independent set of regular activity groups similar to that in Fig 5, except that the frequencies on the activities have changed: Since there is only a single population, the frequencies are not associated with subpopulations any more. Let $x_j^*$ be the optimal frequency for an individual to participate in activity $j \in V_i^*$ for the $i$th regular activity group in the selected activities. Then, $\sum_{j \in V_i^*} x_j^* = \lambda^* m_i / \alpha_i / (r_i + 1)$. The simulation was again run for 10 times with 10 different randomly generated initial strategies for all the individuals. In each time, it converged to an optimal strategy with all frequency values approximating well to their theoretical values. (The simulation results for either single or multiple subpopulations in the above university town are all stored in S3 and S4 Files).

## Discussion

Work on social distancing for mitigating the spread of epidemics has been done in the past [1–4, 21–26], and been intensified after the outbreak of the COVID-19 pandemic [5–8, 27–35]. Social distancing has been and will be one of the most important measures and often the only available one for fighting epidemics and saving lives [1–8]. However, so far it is still very confusing on how social distancing should be practiced or managed when the issue goes beyond six feet. The debate on the issue often leads to a heated political argument especially when it comes to restricting daily lives and businesses and regulating everyone's otherwise normal social activities [13–18].

This paper is to provide a theoretical basis for the understanding of the scientific nature of social distancing and to motivate further investigation into the issue. The points to make in this paper are essentially the following: First, social distancing can be considered as a social dilemma game. The game is played by every individual against his/her population. Everyone needs to make a decision on how to engage in social distancing, or risk being trapped into a dilemma either exposing to deadly diseases or getting no access to necessary social activities. Second, based on the game theory, an optimal strategy exists for the distancing game to balance between health and social or economic concerns. It can be determined in theory as a Nash equilibrium of the game, whether it is for dependent or independent activities and for single or multi-populations. Third, the collective behaviors of social distancing, though difficult to measure, can be simulated by computer. The simulation results justify the game model, and also suggest that an optimal distancing strategy can in fact be achieved naturally through the collective actions of the individuals in the population. Finally, the game model for social distancing shows the dependency between the individuals and the population. The population depends on every individual to make a sensible distancing decision, which not only benefits himself/herself, but also helps the population to maintain a balanced distribution on the activities, which in turn benefits every individual in the population. The last point is rather philosophical, but is true for many collective social behaviors.

This work is based on the multidisciplinary studies on evolutionary dynamics of social biological populations in the past several decades, including the development of evolutionary game theory [19, 20, 36–44], research on social grouping and cooperation [45–53], collective animal behaviors [54–57], and network-based evolutionary games [58–65]. It is worth noting that it is not new to use game theory in epidemic modeling. Most notably is the work on vaccination, where the behavior for taking or refusing a vaccine can be modeled as a social dilemma game. Bauch and Earn in 2004 first named and studied the vaccination game, and investigated the effect of vaccination on the dynamic changes of infections in a given population [66]. In 2015, Tanimoto [62] applied the idea to other applications, and for vaccination, in particular, further extended it to more realistic populations, where individuals interact only with their neighbors in their social networks. Later on, during the COVID-19 pandemic, a series of

investigations have also been carried out on the vaccination game and related issues [67–71]. Note that most of these studies focus on the vaccination game with the percentage of the vaccinated population as a key parameter for the investigation on the dynamic changes of the infections and in particular, the reach of the herd immunity of the population. They incorporate the game model for vaccination into conventional disease dynamic models such as the SIR model for the study of the disease dynamics.

This paper, on the other hand, investigates a game for social distancing, not for vaccination, and therefore, is focused on social distancing for mitigating the spread of epidemics or pandemics, although it uses a similar evolutionary game theory approach to the problem. To be clear, it is not on the effect of social distancing on the infection dynamics of the population, either. The paper is concerned with the problem of how to manage or practice social distancing and in particular, how to participate in some social activities while avoiding some others so that the social contacts of every individual in the given population can be minimized. This problem is formulated as a population game, with the selection of each social activities as a pure distancing strategy, and the selection of a number of social activities as a mixed strategy. The game reaches an equilibrium when every individual can find an optimal strategy that minimizes his/her social contacts.

While vaccination is the most critical measure for stopping epidemics, social distancing can be as important: A vaccine may take at least several months or even a year to develop and test before it can be distributed. Worse, for worldwide distribution, it may take two or three years to complete. Then, there could be millions of people who have contracted the disease and tens or hundreds of thousands of them who have died before the vaccine becomes fully accessible. Therefore, before a vaccine is developed and becomes available, social distancing can be a critical and sometimes an only available tool to fight epidemics and save lives. Yet, research on this subject has been very limited, and the decision making on social distancing often relies on political or practical instincts rather than scientific judgements. The current practice has not been very successful, often struggling in between either complete shut-down of all businesses or free participation in all social activities, while neither approach is practical and sustainable.

The social dilemma for social distancing is demonstrated with a simple example in the Introduction section. This example is similar in nature to the paradox for vaccination. It has only two pure strategies, quarantine or not, similar to the pure strategies in the vaccination game, vaccinate or not. However, the real problem considered in this paper is much more complex than the simple example. It investigates more realistic distancing games when there are a number of social activities corresponding to a number of pure distancing strategies. The number of social activities (or pure distancing strategies) may range from several tens to hundreds and in practice, could be thousands with different levels of contact values. Then, the real distancing game for reducing social contacts while participating in a minimal amount of social activities is actually an $n$-strategy game between every individual and the population. When $n$ is large, the game is nontrivial to solve, especially when there are also dependencies among the social activities.

The work presented in the paper has three aspects that are beyond the complexity of the example game given in the Introduction section. First, it deals with distancing games with $n$ independent social activities (or strategies). Second, it considers distancing games with $n$ dependent social activities (or strategies) as well. Third, it also solves distancing games with multiple subpopulations. For each case, the method to recognize and compute a solution for the game is given. The solution is rigorously proved and further justified through computer simulation. The latter also shows the potential for the development of a computational tool for the determination of the optimal strategies for social distancing. In addition, a more realistic simulation for a population in a small university town has been carried out. The results are

interesting, providing valuable insights into how social distancing could be practiced or organized in a real-world environment.

In general, there may always be dependencies among a given set of social activities, which can be described by a dependency network. In some sense, it is a special type of social network, but is different from conventional social networks, which usually describe the social relationships among individuals in a given population such as the social networks considered in the studies on the vaccination game. This unique feature of the social distancing game allows the payoff function of the game to be defined in terms of a weighted adjacency matrix for the dependency network. The games with a similar type of construction have been studied in theoretical biology and applied mathematics. For example, Vickers and co-workers in 1980s have conducted a series of investigations on this type of games for the study of genetic mutation and selection [58, 72, 73]; In the same time period, Bomze et al. published a series of papers on the same class of games for the solution of the maximum clique problem in combinatorial optimization [59, 74, 75]. Wang et al. 2019 [64] formulated a similar class of games for modeling social networking, which later on was extended to the study on social distancing as discussed in Wu 2021 [65] as well as in this paper. Research along this line is believed to have direct impacts in science as well as applied mathematics.

Wang et al. 2019 [64] worked on a so-called social networking game where the members of a population select to visit among a set of interconnected social sites so they can maximize their social connections. They showed that an optimal strategy can be obtained by choosing a maximal set of completely connected social sites (a maximal clique), and derived the stability conditions for such strategies. The social distancing game, which is to minimize the social contacts, is opposite to the social networking game. It is therefore not surprising that an optimal strategy for social distancing can be obtained by choosing a maximal set of completely separated social sites (a maximal independent set). Wu 2021 [65] investigated the social distancing game and showed that an optimal distancing strategy can be obtained by choosing a maximal regular set of social sites in general. The latter form a maximal $r$-regular graph where each node connects to $r$ other nodes in the graph [76]. The maximal independent sets, maximal strong matchings, and maximal sets of cycles are all maximal $r$-regular sets [77, 78]. In the current paper, the selection of social sites (or social activities) is extended to a mixed set of regular groups of sites (or activities) of any sizes and degrees. The model is also generalized to multiple as well as single populations with separated or connected social sites (or social activities). A series of computer simulations are also performed to justify the theoretical results and to prove the potential use of a computational tool to achieve the optimal distancing strategies for a given distancing game.

Several issues still need to be clarified: First, the game model discussed in this paper is different from general models for collective behaviors in the sense that it assumes or requires all the individuals to be able to make their own decisions, not simply following the crowds. Each time when deciding a strategy $x$, the individual is assumed to be informed for the current population strategy $y$. The individual can then evaluate the potential contact $p_i(y)$ at each activity $i$. If this value is high, the corresponding frequency $x_i$ is decreased, and otherwise, it is increased. It is not simply changed to the population average $y_i$. Second, it is probably unrealistic to assume that the individuals can always make rational decisions. Sometimes, they may just follow the crowds or make random decisions. Some preliminary simulation tests showed that random decisions did slow down or even change the convergence of the distancing strategies. Research along this line will be further pursued in future efforts. Finally, as an evolutionary game, the evolutionary stability of the optimal strategies for the distancing game is not discussed in this paper. It is partially because it requires more mathematical analysis and it would be better to discuss it separately elsewhere. In addition, it seems that for social distancing, it is

sufficient for the strategies to be weakly stable. Indeed, as shown in this paper, many distancing strategies are not necessarily strictly stable, allowing small perturbations in their frequencies, but they are more flexible and may be more preferred for practical purposes.

In addition to the research contribution, this paper is intended to emphasize the scientific nature of social distancing and increase the public awareness of the importance of active individual engagement. Therefore, to make it more accessible, the paper is presented in a more narrative form to avoid heavy technical details. Efforts have been made to use less mathematical terms until the sections for Methods where more rigorous descriptions and proofs are needed. In the sections for Results, simple examples have been used to illustrate the ideas before more substantial results are discussed. Extra testings and detailed simulation results are all recorded and documented separately in Supporting information.

A case study for distancing in a small university town is performed and discussed in this work, showing the potential use of the game model in practice. However, the case was oversimplified. For example, the actual number of the activities was much larger than considered in the study; the determination of the connectivity among the activities was quite arbitrary; the assignment of the contact values is lack of a general rationale; etc. These are all serious issues yet to be addressed in future research. Nonetheless, based on this work, with a reasonable estimate on these parameters, the model is expected to be applicable to real world distancing problems and to provide an effective analytical tool for the investigation and implementation of social distancing.

## Methods

### Analysis general

The game model for social distancing discussed in this paper belongs to a special class of games called evolutionary games or population games [19, 20]. Assume there is a set of $n$ activities $V = \{1, \ldots, n\}$ for a given population. Let $x = (x_1, \ldots, x_n)^T$ be the strategy of an individual, with $x_i$ being the frequency of the individual to participate in activity $i$, $\sum_{i=1}^{n} x_i = 1$. Let $y = (y_1, \ldots, y_n)^T$ be the strategy of the population, with $y_i$ being the average frequency of the population to participate in activity $i$, $\sum_{i=1}^{n} y_i = 1$. Assume each activity is assigned with a contact value $\alpha_i$ for activity $i$. Then, in activity $i$, the individual has an $x_i$ chance to meet a probable population fraction $y_i$, and be able to make $x_i y_i \alpha_i$ of social contacts. In addition, if activity $j$ is connected with activity $i$, i.e., the individuals in activity $i$ can also have social contacts with those in activity $j$, then, in activity $i$, the individual has an $x_i$ chance to also meet a probable population fraction $y_j$, and be able to make $x_i y_j (\alpha_i + \alpha_j)/2$ of social contacts. Here, the contact value for the interactions between the individuals in activities $i$ and $j$ is assumed to be the average of the two corresponding contact values $\alpha_i$ and $\alpha_j$.

Let $A = \{A_{i,j}: i, j = 1, \ldots, n\}$ be the connectivity matrix for the activities, with $A_{i,i} = 1$ for all $i = 1, \ldots, n$, and $A_{i,j} = 1$ if activities $i$ and $j$ are connected, and $A_{i,j} = 0$ otherwise. Then, in activity $i$, the individual can have social contacts $x_i p_i(y)$, where $p_i(y) = \sum_{j=1}^{n} A_{i,j} y_j (\alpha_i + \alpha_j)/2$. Let $A^\alpha = \{A_{i,j}(\alpha_i + \alpha_j)/2: i, j = 1, \ldots, n\}$. Then, $A^\alpha$ is a weighted connectivity matrix, and $A^\alpha = (AW + WA)/2$, where $W$ is a diagonal matrix with $\alpha = (\alpha_1, \ldots, \alpha_n)^T$ as the diagonal vector. It follows that in all $n$ activities, the individual can have social contacts $\sum_{i=1}^{n} x_i p_i(y) = \sum_{i=1}^{n} x_i \sum_{j=1}^{n} A_{i,j} y_j (\alpha_i + \alpha_j)/2 = \sum_{i,j=1}^{n} x_i A_{i,j}^\alpha y_j = x^T A^\alpha y$. The last formula shows a clear dependency of the social contacts of the individual on the strategy $x$ of the individual and strategy $y$ of the population. Therefore, the social contacts of an individual of strategy $x$ in a population of strategy $y$ can be defined as a function $\pi(x, y) = x^T A^\alpha y$.

Assume that the game is played out by every individual against the population in order to minimize his/her social contacts. Then, given a population of strategy $y$, each individual tries to find a strategy $x$ to minimize $\pi(x, y)$. Once a decision is made, $y$ is also changed, and a new decision is again required, and the game continues. In the end, a strategy $x^*$ is hopefully found that is optimal for everyone, when the population strategy also becomes $x^*$, and the game reaches an equilibrium, i.e., a strategy $x^*$ is found such that

$$\pi(x^*, x^*) \leq \pi(x, x^*), \qquad \text{for all } x \in S$$

where $S = \{x \in R^n : \sum_{i=1}^{n} x_i = 1, \ x_i \geq 0, \ i = 1, \ldots, n\}$ is the set of all possible strategies.

A strategy $x^*$ at equilibrium for the distancing game, called a Nash equilibrium, represents a distancing strategy for the given activities. A sharp distinction among the activities based on this strategy is $x_i^* > 0$ for $i$ in a subset $V^*$ of $V$, but $x_i^* = 0$ for all the others. This means that the individuals will only participate in the activities in $V^*$ while avoiding all the others. Based on the evolutionary game theory, at equilibrium, all $p_i(x^*)$ for $i$ such that $x_i^* > 0$ should be equal to the same value, say $\lambda^*$, and all $p_i(x^*)$ for $i$ such that $x_i^* = 0$ should be greater than or equal to $\lambda^*$. These conditions are necessary and also sufficient for $x^*$ to be an equilibrium strategy, as given in the following theorem.

**Theorem 1**. *Let $V = \{1, \ldots, n\}$ be the set of activities for a given population. Let $A^\alpha$ be the weighted connectivity matrix of the activities. Then, a strategy $x^* \in S$ is a Nash equilibrium of the social distancing game if and only if there is a scalar $\lambda^*$ such that $p_i(x^*) = \lambda^*$ for all $i$ such that $x_i^* > 0$ and $p_i(x^*) \geq \lambda^*$ for all $i$ such that $x_i^* = 0$, where $p_i(x^*) = \sum_{j=1}^{n} A_{i,j}^\alpha x_j^*$.* [19, 20, 65]

## Analysis 1: Maximal independent sets of activities

If all the activities are separated, the connectivity matrix $A^\alpha$ is simply a diagonal matrix with $\alpha$ as the diagonal vector. Assume all activities are selected. Then, $p_i(x^*) = \lambda^*$, i.e, $\alpha_i x_i^* = \lambda^*$ for all $i = 1, \ldots, n$. It follows that $1/\lambda^* = 1/\alpha_1 + \cdots + 1/\alpha_n$ and $x_i^* = \lambda^*/\alpha_i$, and $x^*$ is an optimal strategy by Theorem 1. Note that there is no way to select only a subset of activities, for $p_i(x^*) = 0 < \lambda^*$ for any activity $i$ not selected, violating the inequality conditions stated in Theorem 1.

If the activities do have some dependencies but there is a maximal independent set of activities $V^*$, an optimal strategy $x^*$ can be determined with $x_i^* > 0$ for all $i$ in $V^*$ and $x_i^* = 0$ for all $i$ not in $V^*$. Let $p_i(x^*) = \lambda^*$ for all $i$ in $V^*$. Then, $\alpha_i x_i^* = \lambda^*$ for all $i$ in $V^*$. It follows that $1/\lambda^* = \Sigma_{i \in V^*} 1/\alpha_i$, and $x_i^* = \lambda^*/\alpha_i$ for all $i$ in $V^*$. Assume that $p_i(x^*) \geq \lambda^*$ for all $i$ not in $V^*$. Then, $x^*$ is an optimal strategy by Theorem 1.

## Analysis 2: Maximal dependent sets of activities

In general, if there is a maximal independent set of activity groups $V^*$, with each group being a regular set of activities, an optimal strategy $x^*$ can be determined with $x_i^* > 0$ for all $i$ in $V^*$ and $x_i^* = 0$ for all $i$ not in $V^*$. A regular group of activities is a set of activities, each having the same degree of connectivity, i.e, connecting with the same number of other activities in the group. A maximal strong matching of activities is a maximal independent set of activity groups, with each group being a regular set of two activities of degree 1. The following theorem gives a description for the optimal strategy on a maximal independent set of regular groups of activities.

**Theorem 2**. *Let $V^*$ be a maximal independent set of regular activity groups, $V_i^*$, $i = 1, \ldots, m$. Assume that each activity in $V_i^*$ connects with other $r_i$ activities in $V_i^*$ and all the activities in $V_i^*$ are given the same contact value $\alpha_i$. Let $x_j^* > 0$ for all $j$ in $V^*$ such that $\sum_{j \in V_i^*} x_j^* = \lambda^* m_i/\alpha_i/(r_i +$*

1) *for all $i = 1, \ldots, m$, and $x_j^* = 0$ for all $j$ not in $V^*$, where $1/\lambda^* = \sum_{i=1}^{m} m_i/\alpha_i/(r_i + 1)$ and $m_i = |V_i^*|$, $i = 1, \ldots, m$. Then, if $p_j(x^*) \geq \lambda^*$ for all $j$ not in $V^*$, $x^*$ is an optimal strategy.*

*Proof.* For all $j$ not in $V^*$, set $x_j^* = 0$. For all $j$ in $V^*$, let $p_j(x^*) = \lambda^*$ for some $\lambda^*$. Then, add all the equations for $j$ in $V_i^*$ to obtain $\alpha_i(r_i + 1)\sum_{j \in V_i^*} x_j^* = \lambda^* m_i$. It follows that $\sum_{j \in V_i^*} x_j^* = \lambda^* m_i/\alpha_i/(r_i + 1)$ for all $V_i^*$. Add the latter equations again for all $V_i^*$. Then, $1/\lambda^* = \sum_{i=1}^{m} m_i/\alpha_i/(r_i + 1)$. If $p_j(x^*) \geq \lambda^*$ for all $j$ not in $V^*$, $x^*$ is an optimal strategy by Theorem 1.

## Analysis 3: Strategies for multi-populations

Consider $M$ subpopulations, $P_1, \ldots, P_M$, with $M$ corresponding subsets of activities, $H_1, \ldots, H_M$. Let $H = \{1, \ldots, n\}$ be the set of all the activities. Assume that $\cup_{k=1}^{M} H_k = H$, and for each $k$, $H_k \cap H_l \neq \Phi$ for some $l \neq k$. Let $x^{(k)}$ be the strategy vector for $P_k$ with $x_i^{(k)} \geq 0$ for all $i \in H_k$ and $x_i^{(k)} = 0$ for all $i \notin H_k$, $\sum_{i=1}^{n} x_i^{(k)} = 1$. Let $V^*$ be a maximal independent set of regular activity groups, $V_i^*$, $|V_i^*| = m_i$, $i = 1, \ldots, m$. Assume that each activity in $V_i^*$ connects with other $r_i$ activities in $V_i^*$ and all the activities in $V_i^*$ are given the same contact value $\alpha_i$. In addition, each activity group $V_i^*$ is assumed to belong to one or more activity groups $H_k$ and belong to each entirely. Then, with a proof similar to that for Theorem 2, it is easy to verify that a strategy set $\{x^{*(1)}, \ldots, x^{*(M)}\}$ is optimal if $x_j^{*(1)} = \ldots = x_j^{*(M)} = 0$ for all $j$ not in $V^*$, and $\sum_{j \in V_i^*} \sum_{k=1}^{M} x_j^{*(k)} = \lambda^* m_i/\alpha_i/(r_i + 1)$ for all $V_i^* \subset V^*$, and if $p_j(z) \geq \lambda^*$ for all $j$ not in $V^*$, where $z = \sum_{k=1}^{M} x^{*(k)}$, and $1/\lambda^* = (\sum_{i=1}^{m} m_i/\alpha_i/(r_i + 1))/M$.

## Simulation general

The dynamic behaviors of population games have been modeled by using replicator equations [19, 20, 42, 43]. For a distancing game, the same principle applies by considering the population strategy $y_i$ as a probable population distribution in activity $i$. Let $p = (p_1, \ldots, p_n)^T$ be a vector of contact functions. Then, $p_i(y) = \sum_{j=1}^{n} A_{i,j}^{\alpha} y_j$ represents the amount of social contacts one receives from full participation in activity $i$, while $y^T p(y) = \sum_{i=1}^{n} y_i p_i(y)$ represents the average social contacts one receives with strategy $y$, and a system of replicator equations for social distancing can be written in the following form:

$$\dot{y}_i = y_i(y^T p(y) - p_i(y)), \quad i = 1, \ldots, n,$$

meaning that the changing rate of each population fraction $y_i$ in activity $i$ is proportional to the difference between the average social contact $y^T p(y)$ of the population and the maximum social contact $p_i(y)$ of the individual in activity $i$: If $p_i(y)$ is lower than $y^T p(y)$, $y_i$ increases, and otherwise, it decreases. Based on the evolutionary game theory, the equilibrium strategy for the distancing game should always be a fixed point or equilibrium solution of the above system [19, 20].

The simulation in this work is not done by tracking the solution curves of the replicator equations. Instead, it is implemented more like an agent-based model, with every individual in the population playing the game and making responses as the population evolves to an equilibrium state. However, the rationale behind the decision makings of the individuals is still consistent with the general principle of the replicator equations: In every generation, every individual decides to make some changes on his/her strategies. For each activity $i$, if $p_i(y)$ is lower than $y^T p(y)$, $y_i$ needs to be increased and therefore, if the individual frequency $x_i$ is smaller than the population frequency $y_i$, $x_i$ is increased to $y_i$ as an individual contribution; if $x_i$ is larger than $y_i$, $x_i$ is still increased but not as much, only by half of the difference of $x_i - y_i$ or 1

$- x_i$ whichever is smaller. On the other hand, if $p_i(y)$ is higher than $y^T p(y)$, $y_i$ needs to be decreased and therefore, if the individual frequency $x_i$ is larger than the population frequency $y_i$, $x_i$ is decreased to $y_i$ as an individual cooperation; if $x_i$ is smaller than $y_i$, $x_i$ is still deceased but again not as much, only by half of the difference of $y_i - x_i$ or $x_i - 0$ whichever is smaller. There is no guarantee that such a simulation procedure can truly reveal the dynamic behaviors of the distancing game, but it seems to be more intuitive and realistic, and follows whatever the individuals would or should do when it comes to social distancing. In any case, based on all the test results, the simulation procedure has performed well, produced expected dynamic behaviors, and converged to the equilibrium strategies as predicted by the distancing games as well as the replicator equations.

## Simulation 1: Behaviors with independent sets of activities

A population of $m$ individuals are considered with $n$ independent activities. A contact value is assigned to each activity $i$ with $\alpha_i \geq 1$, $i = 1, \ldots, n$. Since all the activities are independent, $A^\alpha$ is just a diagonal matrix with $\alpha = (\alpha_1, \ldots, \alpha_n)^T$ as the diagonal vector. The simulation is run for $K$ generations. At each generation, a strategy change on $x$ is made for every individual once, followed by an update on the strategy $y$ of the population. The change for the frequency $x_i$ on activity $i$ is based on the difference between the maximum possible contacts one can make in activity $i$ and the average contacts of the population, i.e., between $p_i(y) = \alpha_i y_i$ and $y^T p(y) = \sum_{i=1}^n \alpha_i y_i^2$. In the end of the generation, the root-mean-square-deviation $\sigma$ (RMSD) of the strategy $x$ of each individual deviated from the population strategy $y$ is calculated, where $\sigma^2 = \sum_{i=1}^n (x_i - y_i)^2$. The RMSD of the individual strategy is then averaged over all the individuals. The simulation is run for 10 times, with 10 different sets of randomly generated initial strategies for all the individuals.

## Simulation 2: Behaviors with dependent sets of activities

A population of $m$ individuals are considered with $n$ dependent activities. A contact value is assigned to each activity with $\alpha_i \geq 1$, $i = 1, \ldots, n$. The weighted connectivity matrix $A^\alpha = (AW + WA)/2$, where $W$ is a diagonal matrix with $\alpha = (\alpha_1, \ldots, \alpha_n)^T$ as the diagonal vector, and $A$ is the connectivity matrix of the activities, $A_{i,i} = 1$ for all $i = 1, \ldots, n$, and $A_{i,j} = 1$ if activities $i$ and $j$, $i \neq j$, are connected and $A_{i,j} = 0$ otherwise. The simulation is run for $K$ generations. At each generation, a strategy change on $x$ is made for every individual once, followed by an update on the strategy $y$ of the population. The change for the frequency $x_i$ on activity $i$ is based on the difference between the maximum possible contacts one can make in activity $i$ and the average contacts of the population, i.e., between $p_i(y) = \sum_{j=1}^n A_{i,j}^\alpha y_j$ and $y^T p(y) = \sum_{i,j=1}^n A_{i,j}^\alpha y_i y_j$. In the end of the generation, the root-mean-square-deviation $\sigma$ (RMSD) of the strategy $x$ of each individual deviated from the population strategy $y$ is calculated, where $\sigma^2 = \sum_{i=1}^n (x_i - y_i)^2$. The RMSD of the individual strategy is then averaged over all the individuals. The simulation is run for 10 times, with 10 different sets of randomly generated initial strategies for all the individuals.

## Simulation 3: Behaviors of multi-populations

A population of $m$ individuals is divided into $M$ subpopulations, $P_1, \ldots, P_M$, with a subset $H_k$ of activities associated with $P_k$. Let $H = \{1, \ldots, n\}$ be the set of all the activities. Assume $\cup_{k=1}^M H_k = H$, and for each $k$, $H_k \cap H_l \neq \Phi$ for some $l$. Let $x^{(k)}$ be the strategy vector for an individual in $P_k$. Then, $x_i^{(k)} \geq 0$ for all $i \in H_k$, and $x_i^{(k)} = 0$ for all $i \notin H_k$, $\sum_{i=1}^n x_i^{(k)} = 1$. The simulation is run for $K$ generations. At each generation, a strategy change is made on $x^{(k)}$ once for

every individual in $P_k$. An update on the corresponding population strategy, $y^{(k)}$ for $P_k$, is followed. The change for the frequency $x_i^{(k)}$ on activity $i$ for the individual in $P_k$ when $i \in H_k$ is based on the difference between the maximum possible contacts one can make in activity $i$ and the average contacts of $P_k$, i.e., between $p_i(z) = \sum_{i=1}^{n} A_{i,j}^{\alpha} z_j$ and $[y^{(k)}]^T p(z) = \sum_{i,j=1}^{n} A_{i,j}^{\alpha} y_i^{(k)} z_j$, where $z = \sum_{k=1}^{M} y^{(k)}$. In the end of the generation, the root-mean-square-deviation $\sigma^{(k)}$ (RMSD) of the strategy $x^{(k)}$ of each individual in $P_k$ deviated from the corresponding subpopulation strategy $y^{(k)}$ is calculated, where $[\sigma^{(k)}]^2 = \sum_{i=1}^{n} (x_i^{(k)} - y_i^{(k)})^2$. The RMSD of the individual strategy is then averaged over all the individuals in the corresponding subpopulation. The simulation is run for 10 times, with 10 different sets of randomly generated initial strategies for all the individuals.

## Supporting information

**S1 File. Simulation results 1.** Simulation for a small single population with 10 social activities to form a Petersen's diagram.
(PDF)

**S2 File. Simulation results 2.** Simulation for a small multi-population with 10 social activities to form a Petersen's diagram.
(PDF)

**S3 File. Simulation results 3.** Simulation for a single large population in the small university town.
(PDF)

**S4 File. Simulation results 4.** Simulation for a large multi-population in the small university town.
(PDF)

**S5 File. Simulation code 1.** Code for simulation for a small single population with 10 social activities.
(PDF)

**S6 File. Simulation code 2.** Code for simulation for a small multi-population with 10 social activities.
(PDF)

**S7 File. Simulation code 3.** Code for simulation for a single large population in the small university town.
(PDF)

**S8 File. Simulation code 4.** Code for simulation for a large multi-population in the small university town.
(PDF)

## Acknowledgments

The author would like to thank the academic editor for handing the paper, and thank the anonymous referees for carefully reading the paper and providing valuable comments and suggestions.

All simulation work was done in Matlab. The Matlab code for producing all the results presented in the paper is made available in S5–S8 Files.

## Author Contributions

**Conceptualization:** Zhijun Wu.

**Data curation:** Zhijun Wu.

**Formal analysis:** Zhijun Wu.

**Funding acquisition:** Zhijun Wu.

**Investigation:** Zhijun Wu.

**Methodology:** Zhijun Wu.

**Project administration:** Zhijun Wu.

**Resources:** Zhijun Wu.

**Software:** Zhijun Wu.

**Supervision:** Zhijun Wu.

**Validation:** Zhijun Wu.

**Visualization:** Zhijun Wu.

**Writing – original draft:** Zhijun Wu.

**Writing – review & editing:** Zhijun Wu.

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
