## [Decision Letter · Decision Letter 0]

28 Jun 2021

PONE-D-21-16845

Social distancing is a social dilemma game played by every individual against his/her population

PLOS ONE

Dear Dr. Wu,

Thank you for submitting your manuscript to PLOS ONE. After careful consideration, we feel that it has merit but does not fully meet PLOS ONE’s publication criteria as it currently stands. Therefore, we invite you to submit a revised version of the manuscript that addresses the points raised during the review process.

We look forward to receiving your revised manuscript.

Kind regards,

Jun Tanimoto

Academic Editor

PLOS ONE

Journal Requirements:

Reviewers' comments:

Reviewer's Responses to Questions

**Comments to the Author**

1. Is the manuscript technically sound, and do the data support the conclusions?

Reviewer #1: Partly

2. Has the statistical analysis been performed appropriately and rigorously? 

Reviewer #1: No

3. Have the authors made all data underlying the findings in their manuscript fully available?

Reviewer #1: Yes

4. Is the manuscript presented in an intelligible fashion and written in standard English?

Reviewer #1: Yes

5. Review Comments to the Author

Reviewer #1: This work reports a new concept to represent each individual taking whether rather regressive activity, can be said ‘quarantine’ if being as extreme, or usual activity with no attention on the disease spreading to protest himself as well as others around him. The kernel of the idea relies on multi-player game that was inspired from a primitive 2 by 2 game context which was introduced in Fig. 1.

Although I can respect the author’s aspiration to this study, I cannot help saying that there is quite less amount of new scientific findings in the report from the standpoint of applied mathematics and social physics when we noting the affluent knowledge of evolutionary game theory (EGT) and mathematical epidemiology. Thus, honestly speaking, I cannot embrace a positive impression, although there can be another evaluation if the author’s aspiration and bit deviated concept from the established knowledge would be accounted as positive which might be somehow informative to the audience of PLOS that is one of the top journals dealing with universal issues not so much concerned on a limited and specific issue.

Anyhow, to revised the MS, I would strongly suggest the author to review rich stock of previous studies based on the merging concept, termed with ‘Vaccination Game’, where mathematical epidemiology to quantify the dynamics of a disease spreading on a social network, derived from classical models such as SIR, SEIR and others, is profoundly dovetailed with EGT to quantify the dynamics of individual decision making process; whether or not each agent committing a preventive measure such as social-distancing, pre-emptive vaccination, large scale lock-down maybe called social quarantine and others. The authors should cite some representative work, review those and compare what he reporting on the report. For example; (i) Sociophysics Approach to Epidemics, Springer, 2021, (ii) Effect of information spreading to suppress the disease contagion on the epidemic vaccination game, Chaos, Solitons and Fractals 119, 180-187, 2019, (iii) An evolutionary game modeling to assess the effect of border enforcement measures and socio-economic cost: Export-importation epidemic dynamics Chaos, Solitons & Fractals 146, 110918, 2021, (iv) Evolution of cooperation in social dilemmas under the coexistence of aspiration and imitation mechanisms, Physical Review E 102, 032120, 2020, and others.

6. PLOS authors have the option to publish the peer review history of their article (what does this mean?). If published, this will include your full peer review and any attached files.

Reviewer #1: No

---

## [Author Response · Author response to Decision Letter 0]

15 Jul 2021

Dear Editor,

Thank you so much for handling my paper, and thank the reviewers for taking their precious time to read the paper and provide valuable comments. I have carefully read the literature on “the vaccination game” as suggested by the reviewers. I am sorry for my limited knowledge on related research, and am grateful to the reviewers for pointing it to me. In the revised manuscript, I have made a brief review on the work on “the vaccination game”, and cited a number of recent papers on this subject. I have explained why “the social distancing game” discussed in my paper is a different game and why it is also important to study. The following are my responses to the reviewers’ comments.

------------

Reviewer #1’s Comments:

This work reports a new concept to represent each individual taking whether rather regressive activity, can be said ‘quarantine’ if being as extreme, or usual activity with no attention on the disease spreading to protest himself as well as others around him. The kernel of the idea relies on multi-player game that was inspired from a primitive 2 by 2 game context which was introduced in Fig. 1. Although I can respect the author’s aspiration to this study, I cannot help saying that there is quite less amount of new scientific findings in the report from the standpoint of applied mathematics and social physics when we noting the affluent knowledge of evolutionary game theory (EGT) and mathematical epidemiology. Thus, honestly speaking, I cannot embrace a positive impression, although there can be another evaluation if the author’s aspiration and bit deviated concept from the established knowledge would be accounted as positive which might be somehow informative to the audience of PLOS that is one of the top journals dealing with universal issues not so much concerned on a limited and specific issue.

Anyhow, to revised the MS, I would strongly suggest the author to review rich stock of previous studies based on the merging concept, termed with ‘Vaccination Game’, where mathematical epidemiology to quantify the dynamics of a disease spreading on a social network, derived from classical models such as SIR, SEIR and others, is profoundly dovetailed with EGT to quantify the dynamics of individual decision making process; whether or not each agent committing a preventive measure such as social-distancing, pre-emptive vaccination, large scale lock-down maybe called social quarantine and others. The authors should cite some representative work, review those and compare what he reporting on the report. For example; (i) Sociophysics Approach to Epidemics, Springer, 2021, (ii) Effect of information spreading to suppress the disease contagion on the epidemic vaccination game, Chaos, Solitons and Fractals 119, 180-187, 2019, (iii) An evolutionary game modeling to assess the effect of border enforcement measures and socio-economic cost: Export-importation epidemic dynamics Chaos, Solitons & Fractals 146, 110918, 2021, (iv) Evolution of cooperation in social dilemmas under the coexistence of aspiration and imitation mechanisms, Physical Review E 102, 032120, 2020, and others.

------------

Response to Reviewer #1’s Comments:

Thanks for carefully reading the paper and providing valuable comments! I am sorry for my limited knowledge on epidemic research and related work. I am aware of the work on “the vaccination game”, and should have included some discussions in the paper. I have now read a number of recent papers on this subject and cited them in the revised manuscript. A brief review on the work is added in the discussion section of the paper, followed by an explanation on why “the social distancing game” discussed in this paper is a different game and why it is also important to study.

Indeed, much work has been done on “the vaccination game” in recent years, and especially since the outbreak of the COVID-19 pandemic, and it is also considered as a social dilemma game: Bauch and Earn 2004 first modeled the vaccination problem as a game, and investigated the effect of the proportion of the vaccinated population on the dynamic changes of the infections in the population. In 2015, Tanimoto applied this idea to other applications, and for vaccination, in particular, further extended it to more realistic populations, where individuals interact only with their neighbors in their social networks. Later on, during the COVID-19 pandemic, a series of investigations have also been carried out on the vaccination game and related issues, including Lim and Zhang 2020, Arefin and Tanimoto 2020, Kuga, Tanaka, and Tanimoto 2021, Kabir, Chowdhury, and Tanimoto 2021, etc. Note that most of these studies focus on the vaccination game with the percentage of the vaccinated population as a key parameter to investigate the dynamic changes of the infections and in particular, the reach of the herd immunity of the population. They incorporate the game model for vaccination into conventional disease dynamic models such as the SIR model for the study of the disease dynamics. 

The submitted paper, on the other hand, investigates a so-called “social distancing game”, and therefore, is focused on social distancing for mitigating the spread of epidemics or pandemics, not on vaccination, although it uses a similar evolutionary game theory approach to the problem. To be clear, it is not on the effect of social distancing on the infection dynamics in the population either. The paper is concerned with the problem of how to manage or practice social distancing, and in particular, how to participate in some social activities while avoiding some others so that the social contacts of every individual in the given population can be minimized. This problem is formulated as a population game, with the selection of each social activities as a pure distancing strategy, and with the selection of a number of social activities as a mixed strategy. The game reaches an equilibrium when every individual can find an optimal strategy that minimizes his/her social contacts.

While vaccination is the most critical measure for stopping epidemics, social distancing can be as important: A vaccine may take at least several months or even a year to develop and test before it can be distributed. Worse, for worldwide distribution, it may take two or three years to complete. Then, there could be millions of people who have contracted the disease and tens or hundreds of thousands of them who have died before the vaccine becomes fully accessible. Therefore, before a vaccine is developed and becomes available, social distancing can be a critical and sometimes only available tool to fight epidemics and save lives. Yet, research on this subject has been very limited, and the decision making on social distancing often relies on political or practical instincts rather than scientific judgements. The current practice has not been very successful, often struggling in between either complete shut-down of all businesses or free participation in all social activities, while neither approach is practical and sustainable. 

The social dilemma for social distancing is demonstrated with a simple example in the introduction section of the paper. The latter is similar in nature to the paradox for vaccination. The example has only two pure strategies, quarantine or not, similar to the pure strategies in the vaccination game, vaccinate or not. However, the real problem considered in the submitted paper is much more complex than the given example. It investigates more realistic distancing games when there are a number of social activities corresponding to a number of pure strategies. The number of social activities may range from several tens to hundreds and in practice, could be thousands with different levels of contact values. Then, the distancing problem for reducing social contacts while participating in a minimal amount of social activities is actually an n-strategy game between every individual and the population. When n is large, the game is nontrivial to solve, especially when there are dependencies among the social activities. 

The work presented in this paper has three aspects that are beyond the complexity of the example game given in the introduction section. First, it deals with distancing games with n independent social activities (or strategies). Second, it considers distancing games with n dependent social activities (or strategies) as well. Third, it also solves distancing games with multiple subpopulations. For each case, the method to recognize and compute a solution for the game is given. The solution is rigorously proved and further justified through computer simulation. The latter also suggests the potential for the development of a computational tool for the determination of the optimal strategies for social distancing. In addition, a more realistic simulation for a population in a small university town has been carried out. The results are interesting, providing valuable insights into how social distancing could be practiced or organized in a real-world environment. To my knowledge, all these investigations have not been done before, but I do believe that they are important and even urgent to pursue so we can be better prepared for any future outbreaks of epidemics or pandemics. 

Note that there can be dependencies among a given set of social activities, which can be described by a dependency network. In some sense, it is a special type of social networks, but it is different from conventional social networks, which usually describe the social relationships among individuals in a given population such as the social networks considered in the studies on the vaccination game. This unique feature of the social distancing game allows the payoff function of the game to be defined in terms of a weighted adjacency matrix for the dependency network. The games with a similar type of construction have been studied in theoretical biology and applied mathematics. For example, Vickers and co-workers in 1980s have conducted a series of investigations on this type of games for the study of genetic mutation and selection; In the same period, Bomze et al. published a series of papers on the same class of games for the solution of the maximum clique problem in combinatorial optimization. Wang et al. 2019 formulated a similar class of games for modeling social networking, which later on is extended to the study on social distancing as discussed in Wu 2021 as well as in this submitted paper. All these studies have been published in journals on theoretical biology with direct applications in science and public health and on applied mathematics with substantial mathematical contents (modeling, simulation, justification, etc). Research along this line is believed to have direct impacts in science as well as applied mathematics. 

I sincerely thank the reviewer for pointing the research on the vaccination game to me and for asking critical questions. I hope that the explanations above have addressed all the issues raised by the reviewer and justified the scientific value of the paper. To make these issues clear, I have added several paragraphs in the discussion section of the paper, including (more or less the same as described above) a brief review on the work on the vaccination game, discussions on the focus of this paper on social distancing vs. vaccination, its importance in fighting epidemics or pandemics, its scientific challenges, and its impacts in both science and mathematics (see Revised Manuscript with Track Changes). 

The following references are cited and added in the bibliography list of the paper:

53. Kabir K, Kuga K, Tanimoto J. Effect of information spreading to suppress the disease contagion on the epidemic vaccination game. Chaos, Solitons, and Fractals. 2019; 119: 180-187.

62. Tanimoto J. Fundamentals of evolutionary game theory and its applications. Springer. 2015.

66. Bauch C, Earn D. Vaccination and the theory of games. Proceedings of the National Academy of Sciences. 2004; 101: 13391-13394.

67. Lim W, Zhang P. Herd immunity and a vaccination game: An experimental study. PLoS ONE. 2020; 15: e0232652.

68. Arefin M, Tanimoto J. Evolution of cooperation in social dilemmas under the coexistence of aspiration and imitation mechanisms. Physical Review E. 2020; 102: 032120.

69. Kuga K, Tanaka M, Tanimoto J. Pair approximation model for the vaccination game: predicting the dynamic process of epidemic spread and individual actions against contagion. Proc. R. Soc. A. 2021; 476: 20200769.

70. Kabir K, Chowdhury A, Tanimoto J. An evolutionary game modeling to assess the effect of border enforcement measures and socio-economic cost: Export-importation epidemic dynamics. Chaos, Solitons, and Fractals. 2021; 146: 110918.

71. Tanimoto J. Sociophysics approach to epidemics. Springer. 2021.

72. Vickers G, Cannings C. Patterns of ESS's I. Journal of Theoretical Biology. 1988; 132: 387-408.

73. Cannings C, Vickers G. Patterns of ESS's II. Journal of Theoretical Biology. 1988; 132: 409-420.

74. Pardalos P, Ye Y, Han C. Algorithms for the solution of quadratic knapsack problems. Linear Algebra and Its Applications. 1991; 152: 69-91.

75. Bomze I. Regularity vs. degeneracy in dynamics, games, and optimization: A unified approach to different aspects. SIAM Review. 2002; 44: 394-414.

In addition to the response to the reviewers’ comments and suggestions, the following modifications are also made in the paper:

1. In the caption for Fig 3, the phrase “… a population of 100 individuals with independent activities …” in the first sentence is changed to “… a population of 100 individuals with 10 independent activities …” In the caption for Fig 4, the phrase “… a population of 100 individuals with dependent activities …” in the first sentence is changed to “… a population of 100 individuals with 10 dependent activities …”

2. In page 8, below the Table 3, the following paragraph is added:

“The dynamic behaviors of single or multiple subpopulations demonstrated above are also observed when the dependencies among the activities in Fig 2 are changed. The simulation results are obtained and stored in Supporting Information (supporting files S1 and S2) for either single or multiple subpopulations with 10 social activities to form a Petersen's diagram.”

3. In page 12, in the last paragraph before the discussion section, the following sentence is added: “(The simulation results for either single or multiple subpopulations in the above university town are all stored in supporting files S3 and S4 in Supporting Information).”

4. In page 14, line 542, the phrase “… called a maximal clique …” is changed to “… (a maximal clique) …” In line 546, the phrase “… called a maximal independent set …” is changed to “… (a maximal independent set) …” In line 549, “… a maximal regular set of social sites …” is changed to “… a maximal regular set of social sites in general …” In line 551, the word “circles” is changed to “cycles”.

5. In page 15, in the last paragraph before the methods section, line 590, the sentence “… is not necessarily accurate …” is changed to “… is lack of general rationale …”

6. In the Acknowledgements, the following two paragraphs are added:

“The author would like to thank the academic editor for handing the paper, and thank the anonymous referees for carefully reading the paper and providing valuable comments and suggestions.”

“All simulation work was done in Matlab. The Matlab code for producing all the results presented in the paper is made available in Supporting Information (supporting files S5, S6, S7, S8).”

7. The Support Information is now divided into 8 files. The first 4 files, S1, S2, S3, S4, contain detailed simulation results discussed in the paper. The rest 4 files, S5, S6, S7, S8, contain the Matlab code for producing all the simulation results.

8. In the end of the manuscript, after the bibliography list, the following captions are provided for all the supporting information files:

Supporting Information:

S1 File: Simulation Results 1. Simulation for a small single population with 10 social activities to form a Petersen's diagram.

S2 File: Simulation Results 2. Simulation for a small multi-population with 10 social activities to form a Petersen's diagram.

S3 File: Simulation Results 3. Simulation for a single large population in the small university town. 

S4 File: Simulation Results 4. Simulation for a large multi-population in the small university town.

S5 File: Simulation Code 1. Code for simulation for a small single population with 10 social activities.

S6 File: Simulation Code 2. Code for simulation for a small multi-population with 10 social activities.

S7 File: Simulation Code 3. Code for simulation for a single large population in the small university town. 

S8 File: Simulation Code 4. Code for simulation for a large multi-population in the small university town.

9. All figure citations are changed from “Fig. #” to “Fig #” as required by PLoS ONE. All references are written in PLoS ONE format as well.

---

## [Decision Letter · Decision Letter 1]

19 Jul 2021

Social distancing is a social dilemma game played by every individual against his/her population

PONE-D-21-16845R1

Dear Dr. Wu,

We’re pleased to inform you that your manuscript has been judged scientifically suitable for publication and will be formally accepted for publication once it meets all outstanding technical requirements.

Kind regards,

Jun Tanimoto

Academic Editor

PLOS ONE

Additional Editor Comments (optional):

Reviewers' comments:

Reviewer's Responses to Questions

**Comments to the Author**

1. If the authors have adequately addressed your comments raised in a previous round of review and you feel that this manuscript is now acceptable for publication, you may indicate that here to bypass the “Comments to the Author” section, enter your conflict of interest statement in the “Confidential to Editor” section, and submit your "Accept" recommendation.

Reviewer #1: All comments have been addressed

2. Is the manuscript technically sound, and do the data support the conclusions?

Reviewer #1: Yes

3. Has the statistical analysis been performed appropriately and rigorously? 

Reviewer #1: Yes

4. Have the authors made all data underlying the findings in their manuscript fully available?

Reviewer #1: Yes

5. Is the manuscript presented in an intelligible fashion and written in standard English?

Reviewer #1: Yes

6. Review Comments to the Author

Reviewer #1: The revised MS seems well persuasive to what I gave the authors as suggestions. Thus, now, I can say this would be accepted to PLOS one.

7. PLOS authors have the option to publish the peer review history of their article (what does this mean?). If published, this will include your full peer review and any attached files.

Reviewer #1: No

---

## [Editor Report · Acceptance letter]

22 Jul 2021

PONE-D-21-16845R1 

Social Distancing is a Social Dilemma Game Played by Every Individual against His/Her Population 

Dear Dr. Wu:

I'm pleased to inform you that your manuscript has been deemed suitable for publication in PLOS ONE. Congratulations! Your manuscript is now with our production department. 

Kind regards, 

on behalf of

Prof. Jun Tanimoto 

Academic Editor

PLOS ONE